# Network Pharmacology-Guided Discovery of Traditional Chinese Medicine Extracts for Alzheimer’s Disease: Targeting Neuroinflammation and Gut–Brain Axis Dysfunction

**DOI:** 10.3390/ijms26178545

**Published:** 2025-09-03

**Authors:** Ting Zhang, Sunmin Park

**Affiliations:** 1Key Laboratory of Biological Resources and Ecology of Pamirs Plateau in Xinjiang Uygur Autonomous Region, College of Life and Geographic Sciences, Kashi University, Kashi 844000, China; zt229029729@163.com; 2Department of Bioconvergence, Hoseo University, Asan 31499, Republic of Korea; 3Department of Food and Nutrition, Obesity/Diabetes Research Center, Hoseo University, Asan 31499, Republic of Korea

**Keywords:** neuroinflammation, Alzheimer’s disease, herbal extract, oxidative stress, gut–brain axis

## Abstract

Neuroinflammation plays a central role in the pathogenesis of Alzheimer’s disease (AD), with amyloid-β (Aβ) deposition and neurofibrillary tangles driving both central and peripheral inflammatory responses. This study investigated the neuroprotective and anti-inflammatory effects of *Vitex trifolia* (VT), *Plantago major* (PM), *Apocyni Veneti* Folium (AVF), and *Eucommiae folium* (EF) using network pharmacology and a co-culture model of PC12 neuronal and Caco-2 intestinal epithelial cells. Bioactive compounds were identified via high-performance liquid chromatography (HPLC) and screened with network pharmacology analysis, yielding 27 for VT, 10 for PM, 6 for AVF, and 3 for EF. Molecular docking confirmed strong binding affinities between the key bioactive compounds and AD-related targets. A co-culture system of PC12 neuronal and Caco-2 intestinal epithelial cells was established to evaluate the effects of VT, PM, AVF, and EF extracts (at concentrations of 10 µg/mL, 20 µg/mL, and 50 µg/mL) and donepezil hydrochloride (positive-control) on Aβ_25–35_-induced neurotoxicity and lipopolysaccharide (LPS)-induced intestinal inflammation, to assess cell viability, and effects on oxidative stress, mitochondrial function, and inflammatory markers. The VT, PM, AVF, and EF extracts activated phosphoinositide 3-kinase (PI3K)-Akt-glycogen synthase kinase-3β (GSK-3β) signaling, enhanced phosphorylation of AMP kinase, suggesting inhibition of Aβ accumulation and tau hyperphosphorylation (*p* < 0.05). However, donepezil hydrochloride only enhanced AMPK phosphorylation. The extracts reduced lipid peroxidation and acetylcholinesterase by about 5-fold. JC-1 staining confirmed preserved mitochondrial membrane potential, while hematoxylin and eosin staining indicated improved intestinal barrier integrity (*p* < 0.05). PM and AVF reduced the number of mast cells (*p* < 0.05). In conclusion, these findings highlight the multi-target potential of VT, PM, AVF, and EF in mitigating both neuronal and intestinal inflammation. Their dual regulatory effects on the gut–brain axis suggest promising therapeutic applications in AD through the modulation of central and peripheral immune responses.

## 1. Introduction

Alzheimer’s disease (AD) is one of the most prevalent neurodegenerative disorders, affecting over 50 million people worldwide. It is characterized by progressive cognitive decline, memory impairment, and behavioral disturbances [1,2,3]. The hallmark pathological features of AD include the presence of extracellular β-amyloid (Aβ) plaques, intracellular neurofibrillary tangles composed of hyperphosphorylated tau protein, and persistent neuroinflammation mediated by activated microglia and astrocytes [1,2,3]. Among these, neuroinflammation has emerged as a critical driver of neurodegeneration, with recent evidence suggesting that targeting inflammatory pathways may offer novel therapeutic opportunities for AD prevention and treatment.

The gut–brain axis, a bidirectional communication network connecting the central nervous system with the gastrointestinal tract through neural, hormonal, immune, and metabolic pathways, has gained considerable attention in AD research [4]. Accumulating evidence indicates that intestinal dysbiosis, compromised intestinal barrier function, and subsequent systemic inflammation contribute to neuroinflammation and cognitive decline [5]. Lipopolysaccharide (LPS)-induced intestinal inflammation serves as a validated model to study the gut-derived inflammatory signaling implicated in AD pathogenesis [6]. Co-culture systems utilizing Caco-2 intestinal epithelial cells and PC12 neuronal cells act as a valuable in vitro platform to investigate the gut–brain interactions and the therapeutic potential of bioactive compounds [7].

Current therapeutic management of AD primarily involves the use of acetylcholinesterase inhibitors. They offer limited symptomatic relief, do not modify disease progression, and are associated with significant adverse effects during long-term use [8]. Donepezil hydrochloride is a first-line, FDA-approved acetylcholinesterase (AChE) inhibitor for the symptomatic treatment of mild to moderate Alzheimer’s disease. It is one of the most widely used and clinically established drugs for AD, making it a highly relevant benchmark for evaluating potential therapeutic agents [9]. The complex, multifactorial nature of AD pathology necessitates multi-target therapeutic approaches that can simultaneously address neuroinflammation, oxidative stress, and systemic inflammation [10]. Traditional Chinese medicine (TCM) represents a promising source of therapeutic agents with multiple therapeutic targets. The plant-derived compounds present in TCM exhibit several pleiotropic biological activities, including anti-inflammatory, antioxidant, and gut-modulatory effects [11].

Through systematic screening of the high-throughput experiment- and reference-guided (HERB) database (http://herb.ac.cn/), which encompasses 7263 herbs and their associated targets and diseases, we identified four medicinal plants classified as “medicinal food homologous” with documented safety profiles: *Vitex trifolia* (VT), *Plantago major* (PM), *Apocyni veneti folium* (AVF), and *Eucommiae folium* (EF). These plants are rich in bioactive compounds, including flavonoids, terpenoids, and phenolic acids, and possess diverse pharmacological properties, such as anti-inflammatory, hepatoprotective, and neuroprotective effects. However, their specific mechanisms in Aβ-induced neuroinflammation and LPS-induced intestinal inflammation remain unexplored.

This study integrated network pharmacology with experimental validation to elucidate the multi-target mechanisms of these four herbal extracts in AD. Using the Traditional Chinese Medicine Systems Pharmacology (TCMSP) database, we identified active compounds and predicted their interactions with AD- and inflammation-related targets. We validated these predictions in vitro using a Caco-2/PC12 co-culture system to assess the efficacy of VT, PM, AVF, and EF extracts in modulating Aβ-induced neuroinflammation and LPS-induced intestinal inflammation. Our findings provide mechanistic insights into the therapeutic potential of these traditional medicines as multi-target agents for AD treatment through gut–brain axis modulation.

## 2. Results

### 2.1. Network Pharmacology Results

Bioactive compounds from VT, PM, AVF, and EF were screened using the TCMSP database based on OB and DL inclusion criteria. A total of 27, 10, 6, and 3 active components were identified from VT, PM, AVF, and EF, respectively. Notably, vitetrifolin C in VT, procyanidin B1 in AVF, baicalin and stigmasterol in PM, and quercetin and kaempferol in EF exhibited high OB (>30%) and DL (0.24–0.80) (Appendix A). From 14,423 AD-related targets, the top 250 compounds were selected by relevance and intersected with drug targets, yielding 25 overlapping potential AD-related targets (Figure 1A).

A network depicting “active ingredient–disease–target” interactions was constructed (Figure 1B). The 25 shared target genes (blue ovals) link AD (red triangle) with bioactive compounds (orange rectangles) derived from VT, PM, AVF, and EF (green arrows). The bioactive compounds contributing most to AD-related targets were 10 in VT, 5 in PM, 2 in AVF, and 2 in EF. PPI analysis revealed 25 target proteins with 195 interactions, highlighting AKT1, IL1B, and IL6 as central nodes (Figure 1C,D).

GO enrichment (top 20 terms, *p* < 0.01) indicated involvement in cytokine activity (GO:0005125), receptor–ligand interactions (GO:0048018), and immune signaling pathways (Figure 1E). KEGG pathway enrichment identified 88 pathways, with the top 20 significantly enriched (*p* < 0.00005). These included lipid and atherosclerosis (hsa05417), advanced glycation end-product (AGE)-receptor for AGE (RAGE) signaling in diabetic complications (hsa04933), and fluid shear stress–atherosclerosis pathways (hsa05418) (Figure 1F). These pathway enrichment results provided strong predictive pathways for our subsequent experimental validation. Specifically, we focused on experimentally testing the modulation of the AGE-RAGE and the phosphoinositide 3-kinase (PI3K)-Akt signaling pathway, given their central roles in neuroinflammation, apoptosis, and the cross-talk between peripheral and central inflammation, which aligns with our gut–brain axis investigation.

### 2.2. Validation Through Molecular Docking

The binding energy between bioactive compounds (baicalein, kaempferol, luteolin, quercetin) and core AD targets (AKT1, IL1B, IL6) showed the binding affinity between them. The binding energy indicated favorable binding (≤−9.0 kcal/mol), particularly for AKT1, which exhibited the lowest energies: −9.8 (baicalein), −9.7 (kaempferol), and −9.8 kcal/mol (quercetin) (Figure 2A). Hydrogen bonding analysis showed interactions between Akt1 and baicalein (THR-211, SER-205) (Figure 2B), kaempferol (THR-211, LYS-268) (Figure 2C), and quercetin (THR-211, LYS-268, ILE-290) (Figure 2D), supporting their potential as AD-target modulators.

### 2.3. Phytochemical Composition of Herbal Extracts and Active Constituents

Crude extract yields after freeze-drying were 2.08%, 10.21%, 13.61%, and 11.55% in VT, PM, AVF, and EF, respectively. Polysaccharides and polyphenols were highest in EF (28.5 and 27.6 mg/g dried herb) and followed by AVF (24.0 and 22.8 mg/g dried herb) (Appendix A). Flavonoids were highest in VT (2.5 mg/g dried herb). Using HPLC-DAD, the key bioactive compounds were confirmed: Kaempferol in VT, AVF, and EF 5.66 mg/g, 0.136 mg/g, and 0.042 mg/g; baicalein and quercetin in PM 1.217 mg/g and 0.146 mg/g; luteolin in EF 0.641 mg/g; and quercetin in VT 8.75 mg/g. Their presence was consistent. Their presence was consistent with network pharmacology results (Appendix A).

### 2.4. Neuroprotective Effects and Anti-Inflammatory Activity

Treatment with baicalein, kaempferol, luteolin, and quercetin (2 µM and 5 µM) improved viability, with 2 µM luteolin and quercetin showing the strongest effects in PC12 cells administered with 10 µM Aβ and LPS (*p* < 0.05). VT, PM, and AVF extract treatment at 20 µg/mL also restored viability close to control levels (Figure 3A,B; *p* < 0.05), while at 10 µg/mL and 50 µg/mL, AVF increased cell viability more than the normal control without Aβ and LPS administration, but VT, PM, and EF increased compared to the control with Aβ and LPS administration (Appendix A). In LPS-treated Caco-2 cells, baicalein and quercetin (2 µM) and kaempferol; luteolin; and quercetin (5 µM) significantly improved cell survival (Figure 3B and Appendix A; *p* < 0.05). Positive control, VT, PM, AVF, and EF showed protective effects at both doses (Figure 3A; *p* < 0.05).

Aβ and LPS increased lipid peroxidation by about 5.5-fold and AChE activity in PC12 cells. Pre-treatment with AVF and EF (10 µg/mL) only significantly reduced lipid peroxidation levels. Pre-treatment with positive control, VT, PM, AVF, and EF (20/50 µg/mL) significantly reduced lipid peroxidation and AChE levels as much as the control (Table 1 and Appendix A; *p* < 0.05). Baicalein, kaempferol, and luteolin were similarly effective against lipid peroxidation, while quercetin showed non-significant reductions in Aβ and LPS administered dual cell model (Table 1 and Appendix A). All four bioactive compounds reduced LPS-induced peroxidation significantly and inhibited AChE activity at 2 µM (Table 1; *p* < 0.05), suggesting neuroprotective potential via improved cholinergic function.

### 2.5. Molecular Mechanisms and Mitochondrial Function

Gene expression analysis revealed that AVF (50 µg/mL) significantly downregulated *TNF-α*, *IL-1β*, *IL-6*, and *Tau* and upregulated *BDNF* in PC12 cells (Appendix A; *p* < 0.05). VT and EF also showed significant anti-inflammatory and neurotrophic effects at 20 µg/mL. In addition, the positive control group reduced *TNF-α* and *Tau* at 20 µg/mL, while upregulating *BDNF*. (Figure 4A,B; *p* < 0.05). Individual bioactive compounds reduced *TNF-α* and enhanced *BDNF* expression, with baicalein and luteolin being especially effective (Figure 4C,D). In Caco-2 cells, all extracts reduced *TNF-α*, *IL-1β*, and *IL-6* mRNA expression (*p* < 0.05), with VT, PM, and AVF showing effects comparable to the controls (Figure 4A; *p* < 0.05). These results confirm that VT, PM, and AVF had anti-inflammatory and neurotrophic actions. The significant downregulation of these key pro-inflammatory cytokines (*TNF-α*, *IL-1β*, *IL-6*) directly validates the predicted enrichment of the AGE-RAGE signaling pathway and inflammatory-related GO terms. This demonstrates that the extracts effectively suppress the neuroinflammatory and peripheral inflammatory responses predicted by our network pharmacology approach.

Western blot analysis showed that Aβ-induced GSK-3β activation and LPS-induced apoptosis were modulated via the AKT/AMPK/GSK-3β pathway. AVF (20 µg/mL) phosphorylated AKT and reduced apoptosis significantly (*p* < 0.05). PM, AVF, and EF (20 µg/mL) increased AKT phosphorylation, while the positive control, VT, PM, AVF, and EF (20 µg/mL) enhanced AMPK phosphorylation, and VT, PM, AVF, and EF upregulated the phosphorylation of GSK-3β (Figure 5A,B), suggesting these extracts regulate key neuroinflammatory and apoptotic pathways.

Morphological analysis using H&E staining showed that AVF and EF (20/50 µg/mL) restored LPS-induced cell integrity loss in Caco-2 cells (Figure 6A,B and Appendix A; *p* < 0.05). Toluidine blue staining revealed that positive control, PM, and AVF significantly reduced mast cell numbers (*p* < 0.05), while VT and EF had weaker effects at 20 µg/mL (Figure 6A,B; *p* < 0.05). At 50 µg/mL, all extracts normalized mast cell counts (Appendix A; *p* < 0.05), indicating anti-inflammatory properties. Mitochondrial membrane potential (MMP) assessment showed that Aβ and LPS decreased MMP in co-cultured PC12 and Caco-2 cells, as indicated by reduced JC-1 aggregation. Treatment with positive control, VT, PM, AVF, EF, and bioactive compounds restored MMP in a concentration-dependent manner, reducing JC-1 monomers and increasing aggregate formation (Figure 6C and Appendix A), suggesting mitochondrial protection and anti-apoptotic effects.

## 3. Discussion

The present study demonstrates that VT, PM, AVF, and EF exert significant neuroprotective and anti-inflammatory effects through multi-target mechanisms relevant to AD pathology, with particular emphasis on gut–brain axis modulation. Our integrated approach combining network pharmacology with experimental validation using a co-culture model represents a significant advancement in understanding how traditional Chinese medicines can simultaneously address both central and peripheral inflammatory responses associated with AD [12,13].

The present study applied network pharmacology analysis as a powerful tool for predicting the multi-target mechanisms of the four herbal extracts. While targets such as AKT1, IL-1β, and IL-6 are well-established in AD pathology, the novelty of our findings lies in demonstrating coordinated modulation of these interconnected pathways simultaneously across both neuronal and intestinal compartments. Our network pharmacology analysis served as a powerful predictive tool, with in vitro experiments directly validating these predictions through observed increases in Akt phosphorylation and reductions in IL-1β and IL-6 expression. This bridges the critical gap between computational predictions and measurable biological effects, addressing a major limitation of previous network pharmacology studies that relied solely on computational predictions without experimental validation [14,15].

In the present study, the co-culture approach simultaneously evaluated responses in both PC12 neuronal and Caco-2 intestinal epithelial cells, representing a significant methodological advancement over conventional single-cell-type studies [16,17,18]. This represents a significant methodological advancement over conventional single-cell-type studies. Traditional AD research has largely overlooked the bidirectional communication between peripheral inflammation and neurodegeneration, despite mounting evidence that gut–brain axis dysfunction contributes to AD pathogenesis [12,19]. In most cell-based studies, mast cell activation modulates neuroinflammation. However, it has also been shown to alter tight junctions in the intestinal cells in animal studies [20,21]. Our co-culture model revealed that PM and AVF could simultaneously reduce mast cell activation while maintaining both neuronal viability and intestinal barrier integrity. This dual protective effect has not been previously demonstrated in TCM research and provides novel insights into how these extracts may address the systemic inflammatory component of AD pathology through gut–brain axis modulation.

Our findings significantly expand on previous research on individual flavonoids. While earlier studies demonstrated that luteolin exhibits anti-inflammatory properties in neuronal cells [22], our work reveals that luteolin-rich VT extract provides superior protection by simultaneously modulating neuroinflammation and intestinal barrier function. This dual activity represents a novel mechanism not previously described in AD research. Previous studies on quercetin focused primarily on its antioxidative properties in brain tissue [23]. Our results additionally demonstrate that quercetin contributes to gut–brain axis protection. The 5.5-fold reduction in lipid peroxidation observed in our study is greater than the significant reductions commonly reported in PC12 cell studies [23]. This suggests that the co-culture environment may enhance therapeutic efficacy, potentially through cell–cell interactions. The identification of kaempferol as a key mediator of gut–brain axis regulation represents a novel finding. While previous research established kaempferol’s anti-inflammatory properties [24], our study is novel in demonstrating its specific role in maintaining intestinal barrier integrity while simultaneously protecting neurons from Aβ-induced toxicity. This dual protective mechanism has not been previously reported and suggests kaempferol may be a promising therapeutic target for AD interventions focusing on gut–brain axis modulation.

Our network pharmacology analysis highlighted significant enrichment of the AGE-RAGE signaling pathway, providing critical hypotheses for experimental validation. While previous studies showed that individual compounds like quercetin and luteolin could inhibit RAGE-mediated inflammation [17,25]. Our network pharmacology analysis, which highlighted the significant enrichment of the AGE-RAGE Signaling Pathway (hsa04933), provided a critical hypothesis for our in vitro work. Our research provides novel insights into how TCMs can simultaneously inhibit this pathway in both neuronal and intestinal cells. While earlier work demonstrated that individual compounds like quercetin and luteolin could inhibit RAGE-mediated inflammation [16,26], our study reveals that whole extracts provide superior pathway modulation through multi-component synergy. The comprehensive inhibition of AGE-RAGE-associated inflammatory markers (*TNF-α*, *IL-1β*, *IL-6*), as also predicted by our GO analysis of cytokine activity, observed in our co-culture system contrasts with previous studies that showed partial inhibition in single-cell models. This enhanced efficacy suggests that gut–brain axis interactions may amplify the therapeutic effects of these extracts beyond what would be predicted from isolated cellular studies.

The therapeutic potential of targeting the gut–brain axis in AD is strongly supported by growing in vivo evidence. Recent studies have demonstrated that modulating gut microbiota [27,28], or improving intestinal barrier function [29], can attenuate neuroinflammation and cognitive decline in AD rodent models. Our findings build upon this foundation through several novel contributions. While individual studies demonstrated neuroprotective effects of flavonoids in isolated neuronal cultures, our co-culture system provides the first experimental evidence of simultaneous gut–brain protection by VT, PM, AVF, and EF extracts within a unified model system. This dual-compartment approach bridges the gap between single-cell-type studies and the complex in vivo interactions demonstrated in recent animal models [30]. Previous research on baicalein showed anti-inflammatory effects in microglial cells [31], but our dual-compartment model reveals broader therapeutic mechanisms involving both intestinal barrier protection and neuronal survival. Similarly, while earlier studies demonstrated PI3K/AKT pathway activation by individual herbal compounds in neuronal models [32], we show for the first time that whole herbal extracts can coordinate this pathway across both neuronal and intestinal epithelial compartments.

The translational potential of our findings is supported by emerging in vivo evidence demonstrating therapeutic efficacy of gut–brain axis interventions in AD models. Recent studies have shown that gut microbiota modulation through probiotic supplementation significantly improved cognitive function and reduced neuroinflammation in transgenic AD mice. The multi-target approach demonstrated in our co-culture system aligns with successful in vivo interventions that show greatest efficacy when addressing both peripheral inflammation and central neuroprotection simultaneously, suggesting strong translational potential for these herbal extracts as gut–brain axis therapeutics.

Our findings suggest a paradigm shift from traditional CNS-focused AD therapies toward integrated gut–brain axis interventions. Previous clinical trials of individual bioactive compounds showed limited success, partly because they failed to address the systemic nature of AD pathology [33]. Our demonstration that these herbal extracts can simultaneously modulate central and peripheral inflammation suggests that targeting the gut–brain axis may offer superior therapeutic outcomes compared to single-target approaches. The multi-target approach demonstrated in our co-culture system aligns with successful in vivo interventions that show greatest efficacy when addressing both peripheral inflammation and central neuroprotection simultaneously. This convergence between our mechanistic findings and animal model outcomes suggests strong translational potential for these herbal extracts as gut–brain axis therapeutics

### Limitations and Future Directions

First, while our Caco-2/PC12 co-culture model provides valuable mechanistic insights into gut–brain axis interactions and has been validated in previous studies [34,35], it cannot fully recapitulate in vivo complexity, including gut microbiome interactions and blood–brain barrier function. Future research should investigate how these extracts affect microbial communities and their ability to cross the blood–brain barrier in animal models. Second, our study lacks overall pharmacokinetic characterization and systematic comparison between individual compounds and whole extracts. While our concentrations were based on cytotoxicity screening and literature evidence, optimal therapeutic dosing, bioavailability profiles, and potential synergistic or antagonistic interactions within complex herbal formulations remain undetermined. Future studies should employ combination analysis approaches and detailed pharmacokinetic profiling to optimize therapeutic formulations and establish consistent bioactivity relationships. Finally, our findings represent foundational mechanistic research requiring extensive in vivo validation before clinical translation. The pharmacokinetics, bioavailability, safety profiles, optimal dosing regimens, and potential drug interactions of these herbal extracts must be systematically characterized in animal models before any clinical application can be considered.

In conclusion, our integrated approach demonstrates that VT, PM, AVF, and EF extracts mitigate Aβ-induced neurotoxicity and LPS-induced intestinal inflammation in vitro through multi-target mechanisms involving suppression of neuroinflammatory pathways. Furthermore, this study provides novel evidence that VT, PM, AVF, and EF can simultaneously address both neuronal and intestinal components of AD pathology through gut–brain axis modulation. Our innovative co-culture approach reveals therapeutic mechanisms not previously described in TCM research and demonstrates that network pharmacology predictions can be successfully validated through targeted experimental investigations. The dual protective effects observed in our study suggest that integrated gut–brain axis interventions may offer superior therapeutic outcomes compared to traditional single-target approaches, representing a significant advancement in the development of therapeutic strategies for AD.

## 4. Methods

### 4.1. Network Pharmacology Analysis

**Screening of Bioactive Compounds in the Herbs:** Active ingredients from VT, PM, AVF, and EF were screened using the Traditional Chinese Medicine Systems Pharmacology (TCMSP) database (https://www.tcmsp-e.com/#/home, 23 July 2024). Compounds with oral bioavailability (OB) ≥ 30% and drug-likeness (DL) ≥0.18 were selected as bioactive candidates [36]. **Target Identification and Network Construction:** Compound-related targets were identified from the TCMSP database. Target names were standardized using the UniProt database. AD-associated targets were obtained from the GeneCards^®^ database using “Alzheimer’s disease” as the keyword. Common targets between the active ingredients and AD were identified using the VENNY 2.1 drawing tool and visualized with the Cytoscape 3.8.2 application to construct a drug-component-disease-target network. **Protein–Protein Interaction (PPI) Network**: PPI networks were generated using the Search Tool for the Retrieval of Interacting Genes/Proteins (STRING) database with *Homo sapiens* as the organism and a confidence threshold of ≥0.40. The network was analyzed with Cytoscape 3.8.2, and key targets with degree values higher than average were selected for further analysis. **Enrichment Analysis**: Drug-disease overlapping targets were analyzed using the Kyoto Encyclopedia of Genes and Genomes (KEGG) database to identify associated canonical pathways. Gene Ontology (GO) enrichment analysis was performed using the Bioconductor package to explore biological processes, cellular components, and molecular functions related to the identified targets [37]. Significance was set at *p* < 0.05.

### 4.2. Molecular Docking

The chemical structures of key bioactive compounds (baicalein, kaempferol, luteolin, and quercetin) were downloaded from PubChem and prepared using Chem3D 17.0. Target proteins serine/threonine kinase AKT (AKT1, PDB ID: 3o96), interleukin-6 (IL-6, PDB ID: 1ALU), and IL-1β (PDB ID: 1hib) were retrieved from the Research Collaboratory for Structural Bioinformatics (RCSB) Protein Data Bank. Molecular docking was performed using AutoDockTools 1.5.6 with semi-flexible docking parameters. The grid was set to 126 Å × 126 Å × 126 Å centered on the ligand, with 1000 docking runs [38]. The grid box was made of the active site of the protein, corresponding to the website (https://proteins.plus/, 30 July 2024). Binding energies were calculated to evaluate ligand–target interactions, and the lowest-energy conformations were selected for interpretation.

### 4.3. Preparation of Herbal Extract and Measurement of Index Bioactive Compounds

VT, PM, AVF, and EF were purchased from the Korean Oriental Medicine Cooperative (Seoul, Republic of Korea). Each of them was decocted in distilled water (1:10 ratio) at 95 °C for 6 h, filtered, and the process was repeated. Each filtrate was concentrated using rotary vacuum evaporation (Buchi R-215 Rotavapor) at 55 °C until the volume was reduced to approximately 1/10th of the original. They were centrifuged at 11,300× *g* at 4 °C for 20 min, and the supernatant was filtered and freeze-dried. Bioactive compounds (baicalein, kaempferol, luteolin, and quercetin) purchased from the Avention Research Center were used as standards.

Lyophilized herbal extracts were dissolved in methanol (0.1 g/mL) and filtered with a 0.45 μm microporous membrane. Standards such as baicalein, kaempferol, luteolin, and quercetin were also dissolved in methanol (50 μg/mL). High-performance liquid chromatography-diode-array detection (HPLC-DAD) equipped with a SunFire C18 column (4.6 mm × 75 mm, 5 µm) was used to identify the amounts of the index bioactive compounds in the herbal extracts with a flow rate of 1 mL/min, a column temperature of 30 °C, and an injection volume of 10 µL. The chromatographic conditions were optimized. The mobile phase used for analyte separation comprised a 0.1% phosphoric acid solution (A) and acetonitrile (B). The run-time was 4 min, and elution was performed in the isocratic mode with A: 40% and B: 60%. Detection was carried out at 370 nm.

### 4.4. Co-Cell Culture and Cytotoxicity Assessment

An in vitro gut–brain axis model was established using a co-culture system of PC12 neuronal cells and Caco-2 intestinal epithelial cells (KCLB 21721, Korean Cell Line Bank, Seoul, Republic of Korea), as illustrated in Appendix A. PC12 cells were cultured in poly-D-lysine-coated flasks with high-glucose Dulbecco’s Modified Eagle Medium (DMEM, 4.5 g/L; ThermoFisher, Waltham, MA, USA) supplemented with 10% horse serum and 5% fetal bovine serum (FBS). Caco-2 cells were maintained in high-glucose DMEM containing 10% FBS, 0.5% penicillin/streptomycin, and 1% GlutaMAX. All cultures were maintained at 37 °C in a humidified 5% CO_2_ atmosphere with media replacement every 48 h.

Cell Differentiation Protocol: PC12 cells were seeded at 4 × 10^4^ cells/well in poly-D-lysine-coated 96-well plates and differentiated over 7 days using serum-reduced DMEM (1% horse serum, 1% penicillin/streptomycin) supplemented with 50 ng/mL nerve growth factor (NGF; ThermoFisher), with media refreshed every 48 h. Caco-2 cells were seeded at 2.5 × 10^4^ cells/well onto 0.4 μm polyethylene terephthalate (PET) semipermeable membrane inserts (SPLInsert™, Pocheon, South Korea) and cultured for 21 days to establish a polarized intestinal epithelial monolayer with developed microvilli and tight junctions, confirmed by transepithelial electrical resistance (TEER) measurements >300 Ω·cm^2^.

Co-Culture and Treatment Protocol: The experimental timeline followed a standardized 23-day protocol: Days 0–3: Cell expansion and maintenance in 75 cm^2^ culture flasks; Day 3: Caco-2 cell seeding onto collagen type-I precoated inserts (2.5 × 10^4^ cells/well); Days 3–21: Caco-2 differentiation with medium changes every 48 h; Days 5–10: PC12 neuronal differentiation with 50 ng/mL NGF treatment; Day 10: Differentiated PC12 cell seeding onto culture plates; Day 21: Co-culture establishment with 24 h equilibration period; Day 22: Sequential treatment application—first, inflammatory challenge with 10 μM Aβ_25–35_ (Sigma-Aldrich, St. Louise, MO, USA) added to the basolateral chamber (PC12 side) and 10 μM LPS (E. coli O111:B4; Sigma-Aldrich) added to the apical chamber (Caco-2 side) for 24 h, immediately followed by treatment intervention with herbal extracts (10, 20, or 50 μg/mL) or bioactive compounds (2 or 5 μM); Day 23: Sample collection and analysis.

Quality Control and Validation: Cell confluence was monitored daily using phase-contrast microscopy. Caco-2 cells were used when reaching >90% confluence with visible tight junction formation. PC12 differentiation was confirmed by neurite outgrowth assessment, with cells showing neurites ≥2 times the cell body diameter considered adequately differentiated. TEER was measured for Caco-2 monolayers to confirm barrier integrity (>300 Ω·cm^2^) by TEER voltohmmeter (Merck, Darmstadt, Germany), before co-culture establishment. Mycoplasma contamination was routinely tested using PCR-based detection.

Experimental Groups and Controls: Each experiment included the following systematically controlled groups: (1) Normal Control: Vehicle-treated co-cultures (PBS for extracts, DMSO ≤ 0.1% *v*/*v* for compounds); (2) Control: Aβ_25–35_ + LPS + vehicle; (3) Treatment Groups: Aβ_25–35_ + LPS + test compounds at specified concentrations; (4) Positive Control: Aβ_25–35_ + LPS + donepezil (10 μM; Sigma-Aldrich), which was selected as a positive control due to its clinical relevance as a first-line treatment for AD and its specific mechanism of action as a potent AChE inhibitor [39]. It serves as a gold-standard reference for evaluating the efficacy of our extracts against a clinically established benchmark. Preliminary validation confirmed that Aβ_25–35_ and LPS administration (10 μM each) significantly reduced PC12 and Caco-2 cell viability to <50% compared to untreated controls (Appendix A).

Cell viability was assessed using the 3-(4,5-dimethylthiazol-2-yl)-2,5-diphenyltetrazolium bromide (MTT) assay with systematic dose-response screening. Cells were seeded in 96-well plates (n = 4 technical replicates per condition) and allowed to adhere for 24 h before treatment. For systematic preliminary dose-response screening, herbal extracts were tested at concentrations of 10, 20, 50, 100, 150, and 200 µg/mL, while bioactive compounds were tested at 0.1, 0.2, 0.5, 1, 2, and 5 µM to establish the maximum tolerated concentrations and identify the optimal therapeutic window. After 24 h of treatment, cells were incubated with MTT solution (2 mg/mL in PBS; Sigma-Aldrich) for 3 h at 37 °C in a 5% CO_2_ atmosphere. Formazan crystals solubilized with DMSO were measured by absorbance at 590 nm using a microplate spectrophotometer. Cell viability was calculated as a percentage relative to vehicle-treated controls.

Final treatment concentrations of herbal extracts (10, 20, and 50 μg/mL) and bioactive compounds (2 and 5 μM) were selected using three criteria: (1) maintained cell viability ≥85% relative to vehicle controls, indicating minimal cytotoxicity; (2) demonstrated protective efficacy against Aβ23-35/LPS-induced cellular damage in preliminary studies; and (3) physiological relevance based on reported achievable plasma concentrations and established dosing ranges in phytochemical literature [40]. This systematic approach ensured assessment of dose-dependent biological responses within the non-toxic, therapeutically active window.

### 4.5. TBARS and AChE Activity Assays

To assess oxidative stress and cholinergic activity, TBARS and AChE activity were measured post-treatment (n = 4). Lipid peroxidation was quantified using a TBARS assay kit (DoGenBio, Seoul, Republic of Korea), and AChE activity was determined in PC12 cells using a colorimetric AChE assay kit, following the manufacturer’s protocols.

### 4.6. Molecular Analysis: Real-Time PCR Analysis and Western Blot Analysis

Total RNA was extracted from co-cultured cells using the TRIzol reagent (Thermo Scientific Inc., Santa Clara, CA, USA) (n = 4). RNA concentration was determined spectrophotometrically (260/280 nm). One microgram of RNA was reverse-transcribed into cDNA using the RevertAid RT Kit (Thermo Scientific Inc). The diluted cDNA was mixed with primers for the genes of interest (Appendix A) and SYBR™ Green Master Mix (Affymetrix, Santa Clara, CA, USA). The mixture was run in real-time polymerase chain reaction (PCR). mRNA expression of *TNF-α*, *IL-1β*, *IL-6*, *BDNF*, and *Tau* was analyzed using the ΔΔCT method.

After drug treatment, proteins were extracted from co-cultured PC12/Caco-2 cells using a lysis buffer. Protein concentration was determined using the Bio-Rad Protein Assay Kit. Lysates were centrifuged in the presence of protease/phosphatase inhibitors, and 50 µg protein was resolved via sodium dodecyl sulfate-polyacrylamide gel electrophoresis (SDS-PAGE), transferred to polyvinylidene fluoride (PVDF) membranes, and blocked with 3% bovine serum albumin (BSA). Membranes were incubated overnight at 4 °C with primary antibodies (β-actin, protein kinase B (Akt), phosphorylated (p)-Akt (Ser473), 5′ AMP-activated protein kinase (AMPKα), p-AMPKα (Thr172), glycogen synthase kinase (GSK)-3β, p-GSK-3β (Ser9)), followed by Alexa Fluor 700-conjugated secondary antibodies (1:20,000; Cell Signaling Technology, Danvers, MA, USA) [41]. Detection was performed using the Immun-Star™ WesternC™ Chemiluminescence Kit.

### 4.7. Assessment of Cellular Morphology and Mitochondrial Function

Caco-2 cells grown on Transwell inserts were fixed with 4% formalin after treatment, then stained using hematoxylin–eosin (H&E) and toluidine blue (Sigma-Aldrich, St. Louis, MO, USA) (n = 4), following a modified protocol based on the method elucidated by Yoshimoto et al. [42]. Stained cells were examined under an optical microscope at 100× magnification. Mitochondrial membrane potential (MMP) was assessed using the JC-1 dye (2 µM; MMP Assay Kit), following the manufacturer’s instructions and the Meng et al. method [40]. After staining for 30 min in a CO_2_ incubator, cells were washed and visualized using an inverted fluorescence microscope. Red and green fluorescence intensities represented polarized and depolarized mitochondria, respectively.

### 4.8. Statistical Analysis

All experiments were performed with n = 4 technical replicates per experimental condition and repeated across multiple independent experiments to ensure reproducibility. Data are presented as mean ± standard deviation (SD). Statistical significance was determined using one-way ANOVA followed by Tukey’s post hoc multiple comparison test, conducted using SPSS 16.0 (IBM) software. For multi-factorial experimental designs, two-way ANOVA was employed where appropriate. Different lowercase letters (a, b, c, d) above bars indicate statistically significant differences between groups (*p* < 0.05). All statistical analyses included appropriate control groups, and *p*-values were adjusted for multiple comparisons when necessary.

## Figures and Tables

**Figure 1 ijms-26-08545-f001:**
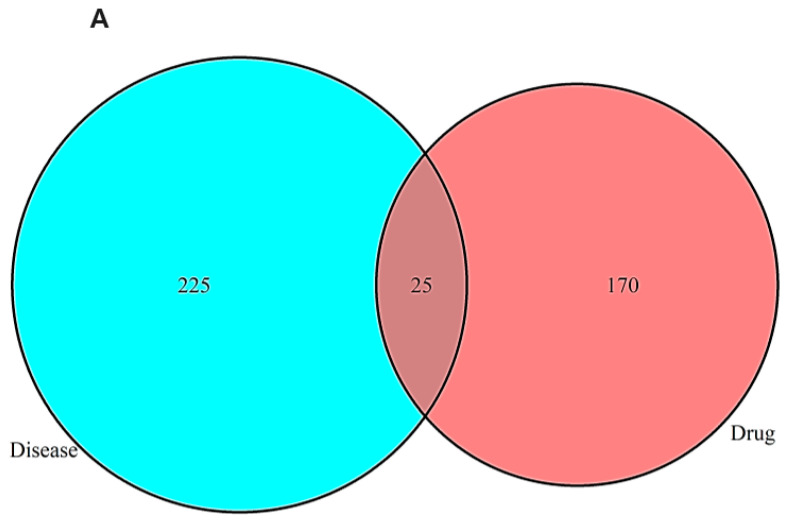
Network pharmacology analysis of TCM active constituents (VT, PM, AVF, EF) and their interactions with AD-related targets. (**A**) Venn diagram analysis showing the overlap and cross-talk between target genes of VT, PM, AVF, EF, and AD. (**B**) Network topology diagram illustrating the relationships between TCM active ingredients (VT, PM, AVF, EF) and AD molecular targets. Blue ovals represent AD-related molecular targets, red triangles represent diseases, green diamonds represent the four TCM compounds (VT, PM, AVF, EF), orange rectangles represent active ingredients, and connecting lines represent interactions between TCM active ingredients and their molecular targets. The network demonstrates the multi-target therapeutic potential of the TCM compounds against AD. (**C**) Protein–protein interaction (PPI) network data retrieved from the STRING database, showing detailed molecular interactions among the identified targets. (**D**) Constructed PPI network for the combined targets of VT, PM, AVF, EF, and AD. Node size and color corresponds to degree centrality (DC) values, with larger and blue color nodes indicating higher connectivity. Edges represent interactions between molecular targets. (**E**) Bubble chart showing Gene Ontology (GO) functional enrichment analysis of core targets for VT, PM, AVF, EF, and AD. The x-axis represents the number of enriched genes, the y-axis lists the functional categories, bubble size indicates enrichment significance, and color intensity represents the degree of enrichment (redder colors indicate higher enrichment and lower *p*-values). (**F**) Bubble chart displaying KEGG pathway enrichment analysis of core targets for VT, PM, AVF, EF, and AD. Red-highlighted pathways represent critical molecular targets for the anti-AD therapeutic effects of VT, PM, AVF, and EF. The x-axis shows the number of enriched genes, the y-axis lists pathway names, and bubble characteristics (size and color) indicate enrichment significance and degree (larger, redder bubbles represent higher enrichment and lower *p*-values). Network pharmacology analysis was performed using Cytoscape 3.8.2 (http://www.cytoscape.org; 12 August 2024). KEGG pathway analysis was conducted using the KEGG database from Kanehisa Laboratories, Kyoto University, Japan (https://www.kegg.jp/; 21 August 2024). Abbreviations: AD, Alzheimer’s disease; AVF, Apocyni *Veneti Folium*; DC, degree centrality; EF, *Eucommiae folium*; GO, Gene Ontology; KEGG, Kyoto Encyclopedia of Genes and Genomes; PM, *Plantago major*; PPI, protein–protein interaction; STRING, Search Tool for the Retrieval of Interacting Genes/Proteins; TCM, traditional Chinese medicine; VT, *Vitex trifolia*.

**Figure 2 ijms-26-08545-f002:**
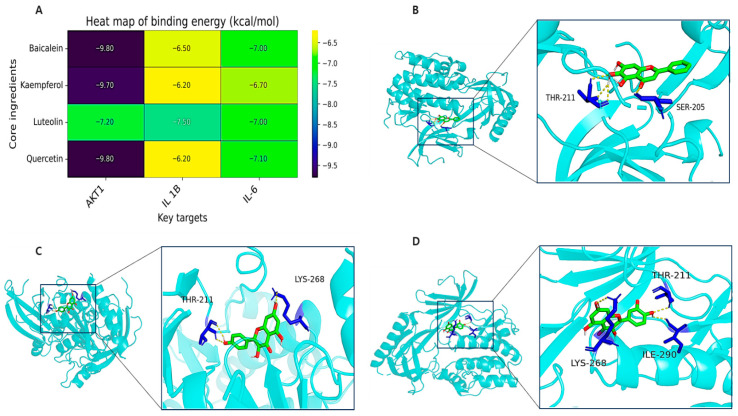
Molecular docking analysis of key target proteins with core active ingredients. (**A**) Heatmap showing binding energies (kcal/mol) between the top three key target proteins and four core active ingredients. Binding energies ≤ −9.0 kcal/mol indicate favorable binding interactions. (**B**) Molecular docking visualization of Akt1 protein in complex with baicalein, showing the overall protein–ligand binding conformation and detailed 3D interactions between the ligand and surrounding amino acid residues. (**C**) Molecular docking visualization of Akt1 protein in complex with kaempferol, displaying the binding mode and specific amino acid interactions within the active site. (**D**) Molecular docking visualization of Akt1 protein in complex with quercetin, illustrating the binding conformation and molecular interactions with key residues. All three compounds (baicalein, kaempferol, and quercetin) demonstrated strong binding affinity to Akt1 with binding energies < −9.0 kcal/mol, suggesting favorable therapeutic interactions.

**Figure 3 ijms-26-08545-f003:**
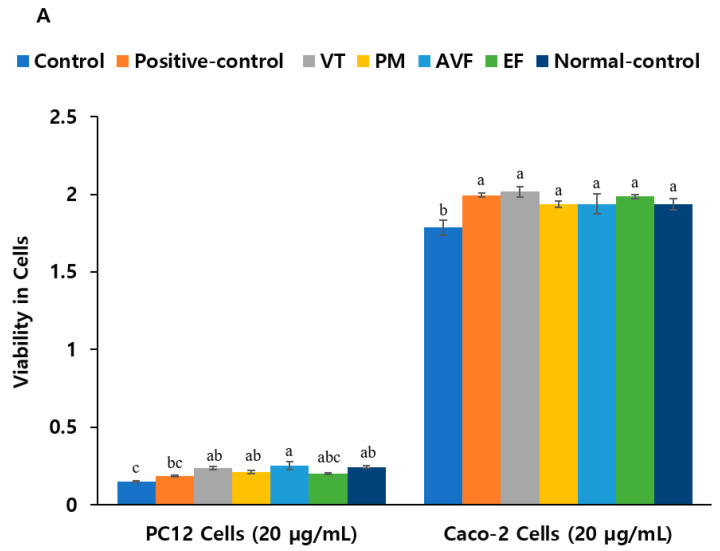
Cell viability assessment of PC12/Caco-2 co-cultures treated with TCM extracts and active compounds. (**A**) Cell viability of PC12/Caco-2 co-cultures exposed to Aβ and LPS damage, followed by treatment with 20 μg/mL concentrations of VT, PM, AVF, and EF extracts. (**B**) Cell viability of PC12/Caco-2 co-cultures exposed to Aβ and LPS damage, followed by treatment with 2 μM concentrations of bioactive compounds. PC12 cells were initially stimulated with NGF to enhance neuronal differentiation (1.5–2-fold increase in proliferation). Co-cultured cells were then pre-treated with 10 μM Aβ or LPS for 1 h to induce cytotoxicity. Subsequently, PC12 cells (basal layer) were treated with herbal extracts (10, 20, 50 μg/mL) or bioactive compounds (2, 5 μM) in low NGF medium, while Caco-2 cells (upper layer) received the same concentrations of LPS treatment. Donepezil hydrochloride served as positive control. After 24 h treatment, cell viability was assessed using the MTT assay. Data are presented as mean values. Different superscript letters (a, b, c) indicate significant differences between groups within each parameter (Tukey’s test, *p* < 0.05). Abbreviations: Aβ, β-amyloid; AVF, *Apocyni Veneti Folium*; EF, *Eucommiae folium*; LPS, lipopolysaccharide; MTT, 3-(4,5-dimethylthiazol-2-yl)-2,5-diphenyltetrazolium bromide; NGF, nerve growth factor; PM, *Plantago major*; TCM, traditional Chinese medicine; VT, *Vitex trifolia*.

**Figure 4 ijms-26-08545-f004:**
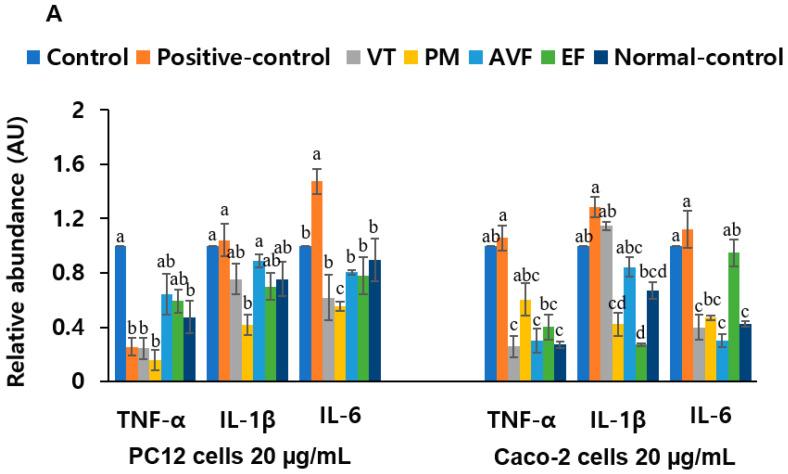
Relative mRNA expression of PC12 cells/Caco-2 cells co-culture. (**A**) Inflammatory cytokine mRNA expression (*TNF-α*, *IL-1β*, *IL-6*) in PC12 and Caco-2 cells following treatment with 20 μg/mL concentrations of VT, PM, AVF, and EF extracts. (**B**) Neurodegeneration-related mRNA expression (*Tau* and *BDNF*) in PC12 cells following treatment with 20 μg/mL concentrations of VT, PM, AVF, and EF extracts. (**C**) Inflammatory cytokine mRNA expression (*TNF-α*, *IL-1β*, *IL-6*) in PC12 and Caco-2 cells following treatment with 2 μM concentrations of active ingredients. (**D**) Neurodegeneration-related mRNA expression (*Tau* and *BDNF*) in PC12 cells following treatment with 2 μM concentrations of active ingredients. PC12 and Caco-2 cells were maintained in co-culture. PC12 cells were exposed to 10 μM Aβ for 1 h to induce neurodegeneration, while Caco-2 cells were exposed to 10 μM LPS for 1 h to induce intestinal inflammation. Following damage induction, cells were treated for 24 h with vehicle control (cell water), donepezil hydrochloride (positive control), TCM extracts (VT, PM, AVF, EF), or bioactive compounds. mRNA was then extracted from both cell types for gene expression analysis. Data are presented as mean ± standard error. Different letters (a, b, c, d) above bars indicate significant differences between treatment groups (Tukey’s test, *p* < 0.05). Identical letters indicate no significant difference. Abbreviations: Aβ, β-amyloid; AVF, *Apocyni Veneti Folium*; BDNF, brain-derived neurotrophic factor; EF, *Eucommiae folium*; IL-1β, interleukin-1β; IL-6, interleukin-6; LPS, lipopolysaccharide; PM, *Plantago major*; TCM, traditional Chinese medicine; TNF-α, tumor necrosis factor-α; VT, *Vitex trifolia*.

**Figure 5 ijms-26-08545-f005:**
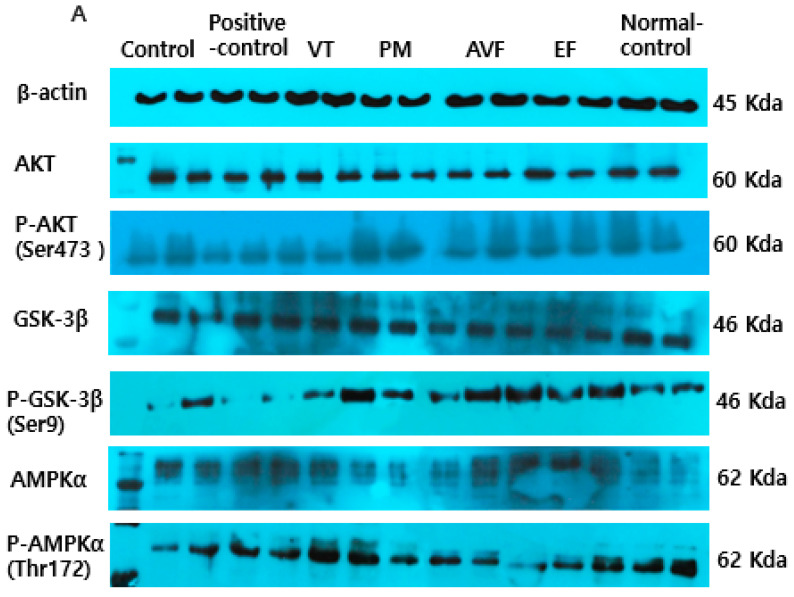
Insulin signaling pathway in PC12 cells by Western blot. (**A**) Representative Western blot images showing protein expression levels of key insulin signaling pathway components (Akt, GSK-3β, AMPK) and their phosphorylated forms in PC12 cells from co-culture experiments. (**B**) Quantitative analysis of total and phosphorylated protein levels of Akt, GSK-3β, and AMPK in PC12 cells following treatment with TCM extracts. Experimental protocol: PC12 and Caco-2 cells were maintained in co-culture with PC12 cells in the basal layer and Caco-2 cells in the upper layer. PC12 cells were exposed to 10 μM Aβ for 1 h to induce neuronal damage, while Caco-2 cells were simultaneously exposed to 10 μM LPS for 1 h to simulate intestinal inflammation. Following damage induction, cells were treated for 24 h with vehicle control, donepezil hydrochloride (positive control), or 20 μg/mL concentrations of VT, PM, AVF, and EF extracts. PC12 cells were then harvested for Western blot analysis of insulin signaling pathway proteins, including total and phosphorylated forms of Akt, AMPK, and GSK-3β, with β-actin serving as loading control. Data are presented as mean ± standard error. Different letters (a, b, c) above bars indicate significant differences between treatment groups (Tukey’s test, *p* < 0.05). Identical letters indicate no significant difference. Abbreviations: Akt, AKT serine/threonine kinase; AMPK, AMP-activated protein kinase; AVF, *Apocyni Veneti Folium*; EF, *Eucommiae folium*; GSK-3β, glycogen synthase kinase-3β; LPS, lipopolysaccharide; PM, *Plantago major*; TCM, traditional Chinese medicine; VT, *Vitex trifolia*; Aβ, β-amyloid.

**Figure 6 ijms-26-08545-f006:**
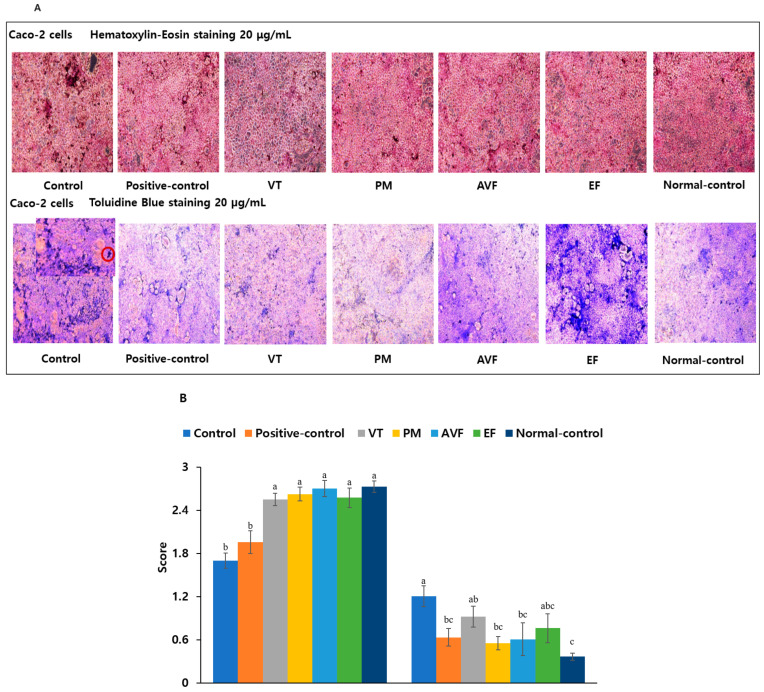
Morphological and mitochondrial analysis of PC12/Caco-2 co-cultures. (**A**) Hematoxylin and eosin (H&E) staining of Caco-2 cells following treatment with 20 μg/mL TCM extracts in co-culture experiments (100× magnification). (**B**) Toluidine blue staining of Caco-2 cells showing mast cell infiltration (indicated by red circles) following treatment with 20 μg/mL TCM extracts in co-culture experiments (100× magnification). (**C**) JC-1 fluorescent staining of PC12/Caco-2 co-cultures demonstrating mitochondrial membrane potential and cellular protection following treatment with TCM extracts (20 μg/mL) or active ingredients (2 μM) (200× magnification). Experimental protocol: PC12 and Caco-2 cells were maintained in co-culture with PC12 cells in the basal layer and Caco-2 cells in the upper layer. PC12 cells were exposed to 10 μM Aβ for 1 h, while Caco-2 cells were exposed to 10 μM LPS for 1 h to induce cellular damage. Following damage induction, cells were treated for 24 h with vehicle control, donepezil hydrochloride (positive control), VT, PM, AVF, or EF extracts (20 μg/mL), or active ingredients (2 μM). JC-1 fluorescent staining was used to examine the degree of apoptosis in Caco-2 cells and PC12 cells. The mitochondrial membrane potential was expressed as red fluorescence. The more fluorescent red (Caco-2 cells) and fluorescent green (PC12-cells), the higher the intensity of cell protection and the inhibition of cell apoptosis. Data are presented as mean ± standard error. Different letters (a, b, c) above bars indicate significant differences between treatment groups (Tukey’s test, *p* < 0.05). Identical letters indicate no significant difference. Abbreviations: AVF, Apocyni Veneti Folium; EF, Eucommiae folium; H&E, hematoxylin and eosin; JC-1, 5,5′,6,6′-tetrachloro-1,1′,3,3′-tetraethylbenzimidazolylcarbocyanine iodide; LPS, lipopolysaccharide; PM, *Plantago major*; VT, *Vitex trifolia*.

**Table 1 ijms-26-08545-t001:** Caco-2 cells /PC12 cells TBARS test and PC12 cells Acetylcholinesterase (AChE) activity detection.

A. Herbal extracts (20 μg/mL)			
	PC12 Cell TBARs (mg/dL)	Caco-2 Cell TBARs (mg/dL)	PC12 Cell AChE activity (U/mL)
Control	0.98 ± 0.008 ^a^	2.36 ± 0.001 ^a^	4.49 ± 0.022 ^a^
Positive-control	0.12 ± 0.002 ^b^	0.60 ± 0.002 ^b^	0.78 ± 0.019 ^c^
VT	0.24 ± 0.001 ^b^	0.76 ± 0.002 ^b^	2.97 ± 0.018 ^b^
PM	0.25 ± 0.004 ^b^	0.43 ± 0.002 ^b^	2.49 ± 0.005 ^b^
AVF	0.17 ± 0.002 ^b^	0.07 ± 0.000 ^c^	2.57 ± 0.024 ^b^
EF	0.26 ± 0.003 ^b^	0.56 ± 0.003 ^b^	2.15 ± 0.133 ^b^
Normal-control	0.13 ± 0.001 ^b^	0.43 ± 0.000 ^b^	2.46 ± 0.016 ^b^
B. Index compounds (2 μM)			
	PC12 Cell TBARs (mg/dL)	Caco-2 Cell TBARs (mg/dL)	PC12 Cell AChE activity (U/mL)
Control	10.16 ± 0.021 ^a^	7.25 ± 0.028 ^a^	4.49 ± 0.023 ^a^
Baicalein	8.45 ± 0.011 ^b^	6.79 ± 0.011 ^b^	3.05 ± 0.033 ^b^
Kaempferol	6.12 ± 0.016 ^c^	6.93 ± 0.008 ^b^	2.46 ± 0.016 ^b^
Luteolin	8.73 ± 0.041 ^b^	6.51 ± 0.012 ^b^	2.87 ± 0.054 ^b^
Quercetin	9.60 ± 0.010 ^ab^	6.87 ± 0.023 ^b^	3.07 ± 0.024 ^b^
Normal-control	6.83 ± 0.000 ^c^	6.64 ± 0.017 ^b^	2.04 ± 0.018 ^b^

Caco-2 cells/PC12 cells TBARS test and PC12 cells AChE activity detection with 20 µg/mL, 2 µM concentrations of positive control, VT, PM, AVF, EF extracts, and active ingredients in co-culture experiments. Caco-2 cells were differentiated for 21 days and co-cultured with PC12 cells for 1 day. The cells were treated with 10 μM Aβ and LPS for 24 h before the TBARs assay or AChE assay. The positive control, VF, PM, AVF, EF extracts, and active ingredients were added to the culture medium 1 h after Aβ_25–35_ and LPS treatment. Data are presented as mean ± standard error. Different letters (a, b, c) above bars indicate significant differences between treatment groups (Tukey’s test, *p* < 0.05). Identical letters indicate no significant difference. Abbreviations: VT, *Vitex trifolia*; PM, *Plantago major*; AVF, *Apocyni Veneti Folium*; EF, *Eucommiae folium*; TBARs, thiobarbituric acid reactants; AChE, acetylcholinesterase.

## Data Availability

The original contributions presented in this study are included in the article/Appendix A. Further inquiries can be directed to the corresponding author(s).

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
