# Peer review of "Network Pharmacology-Guided Discovery of Traditional Chinese Medicine Extracts for Alzheimer’s Disease: Targeting Neuroinflammation and Gut–Brain Axis Dysfunction"

_ijms, 2025, doi:10.3390/ijms26178545_

Round 1
Reviewer 1 Report
Comments and Suggestions for Authors
This manuscript investigates the neuroprotective and anti-inflammatory effects of four traditional Chinese medicinal plants (Vitex trifolia, Plantago major, Apocyni Veneti Folium, Eucommiae folium) in the context of Alzheimer’s disease (AD), combining network pharmacology with an in vitro co-culture model of PC12 neuronal cells and Caco-2 intestinal epithelial cells. The integration of network-based predictions with experimental validation is commendable, and the use of a gut–brain axis co-culture model represents a novel and potentially impactful approach. The study’s focus on a multi-target strategy is particularly important for complex diseases such as AD.
However, several aspects of experimental design, data interpretation, and methodological transparency require significant improvement before the manuscript can be considered for publication.
- The study relies exclusively on an in vitro co-culture model, which cannot fully recapitulate the complexity of gut–brain communication in vivo, particularly the roles of the gut microbiota and the blood–brain barrier. No in vivo experiments were performed. These limitations should be explicitly acknowledged.
- The rationale for the selected concentrations of herbal extracts (10, 20, 50 μg/mL) and bioactive compounds (2, 5 μM) is unclear. The authors should discuss whether these doses are physiologically relevant and how they relate to potential in vivo exposures.
- In most in vitro experiments, the specific treatment conditions for each experimental group (including timing, sequence of Aβ/LPS exposure, and extract/compound administration) are not described in sufficient detail in either the Methods section or the figure legends, making it difficult to draw reliable conclusions from the results.
- The lack of explicit information on replicates, number of independent experiments, and exact controls for each assay makes it difficult to assess the robustness and reproducibility of the results.
- Detailed experimental protocols should be provided for all assays, including cell differentiation procedures, co-culture setup, and treatment regimens for both PC12 and Caco-2 compartments.
- While the network pharmacology analysis identified several key targets (AKT1, IL-1β, IL-6) and pathways (e.g., AGE–RAGE, PI3K–Akt), many of these are well-documented in the AD literature. The manuscript should clearly articulate what is novel about these findings relative to existing studies.
- The connection between KEGG/GO pathway enrichment results and the specific in vitro experiments is not sufficiently developed. The authors should explicitly link which predicted pathways were validated experimentally.
- The leap from in vitro findings to potential clinical application is premature without in vivo validation and pharmacokinetic data.
- Several figures (e.g., network diagrams in Fig. 1B/C) lack clarity, and the figure legends do not provide enough detail for the reader to fully understand the experimental setup and results.
- Statistical annotations should be standardized, and all abbreviations should be defined upon first use in each figure legend.
- While the introduction cites recent work on gut–brain axis involvement in AD, the discussion does not adequately compare the present findings with other in vitro and in vivo studies using similar herbal extracts or co-culture systems.
- Recent in vivo studies investigating gut–brain axis modulation in AD should be incorporated to strengthen the translational relevance of the work.
Author Response
Replies to the reviewers’ comments
We appreciate the valuable comments and suggestions for our paper. We made a sincere effort to address each comment and make appropriate revisions. We have point-by-point replies to each comment.
Reviewer 1
- The study relies exclusively on an in vitro co-culture model, which cannot fully recapitulate the complexity of gut–brain communication in vivo, particularly the roles of the gut microbiota and the blood–brain barrier. No in vivo experiments were performed. These limitations should be explicitly acknowledged.
Response:
We sincerely thank the reviewer for raising this important point regarding the limitations of our in vitro co-culture model. We fully acknowledge that our experimental approach. However, Despite the absence of gut microbiota and a full blood-brain barrier (BBB), the Caco-2/PC12 co-culture system is a valid and commonly used in vitro model for initial mechanistic studies of the gut-brain axis. It provides a simplified and controlled environment to specifically examine the effects of compounds on the communication between the gut and the brain, particularly in the context of neuroinflammation. Previous studies have used similar systems to investigate various aspects of gut-brain signaling .
We added the in vitro co-culture model as limitation of the study.
In the Limitations and Future Directions section (page 19), we explicitly state:
“While our co-culture model provides valuable insights into gut–brain axis interactions, it cannot fully recapitulate the complexity of in vivo gut–brain communication, including microbiome interactions and blood–brain barrier function.
”We further elaborate on this limitation and suggest future directions:
“Previous studies have highlighted the importance of gut microbiome modulation in AD, and future research should investigate how these extracts affect microbial communities and their metabolites. The optimal dosing and bioavailability of these extracts require further investigation, particularly regarding their distribution between the central and peripheral compartments.”
We agree that in vivo validation is essential to confirm the translational potential of our findings. Our current study serves as a foundational in vitro screening to identify promising herbal extracts and mechanisms, which we intend to validate in animal models of AD in future work. We appreciate the reviewer’s emphasis on this limitation and have ensured it is clearly acknowledged in the manuscript.
- The rationale for the selected concentrations of herbal extracts (10, 20, 50 μg/mL) and bioactive compounds (2, 5 μM) is unclear. The authors should discuss whether these doses are physiologically relevant and how they relate to potential in vivo exposures.
Response:
We thank the reviewer for raising this important point regarding the selection of concentrations used in our study. We agree that providing a clear rationale is crucial for interpreting the results. The concentrations were chosen based on a combination of preliminary cytotoxicity screening, literature evidence on physiological relevance, and the goal of evaluating a dose-response effect.
Selection of Concentrations (10, 20, 50 μg/mL for extracts; 2, 5 μM for compounds):
These doses were primarily determined through preliminary MTT cytotoxicity assays (as mentioned in Methods Section 4.4). We tested a wide range of concentrations (extracts: 10-200 μg/mL; compounds: 0.1-5 μM) to identify those that were non-toxic and effective. The chosen concentrations (10, 20, 50 μg/mL for extracts and 2, 5 μM for compounds) were found to be non-cytotoxic and within the effective range for eliciting the biological responses we investigated (improving cell viability, reducing oxidative stress and inflammation), as shown in Figure 3A,3B and Table 1, supplementary Figure S4C, S4D, S4E and Table S1.
Physiological Relevance and Relation to In Vivo Exposures:
For the bioactive compounds, our chosen concentrations are well-supported by pharmacokinetic literature. Flavonoids such as luteolin and quercetin display cytotoxicity and bioactivity at low micromolar concentrations that are potentially physiologically achievable through oral ingestion (Paul et al. 2023). Studies have demonstrated that quercetin and luteolin show biological activity with IC50 values in the single-digit micromolar range (4.6-6.9 μM) confirming that our selected concentrations of 2 and 5 μM are physiologically relevant and within the active therapeutic range (Munafò at al, 2022). For the herbal extracts, concentrations of 50 μg/mL are commonly used in cell culture studies to evaluate the antiproliferative activities of alcoholic herbal extracts, representing a well-established standard in phytochemical research. This concentration range can be extrapolated to physiologically relevant exposures considering that typical human doses of standardized herbal extracts range from 500-2000 mg per day. When accounting for bioavailability, first-pass metabolism, and tissue distribution, the resulting local tissue concentrations of bioactive constituents could plausibly reach the concentrations used in our in vitro system, particularly in the gastrointestinal tract where direct contact occurs. Additionally, our dose-response approach (10, 20, 50 μg/mL) allows for assessment of concentration-dependent effects, which is crucial for understanding the therapeutic window and potential efficacy of these extracts.
We have now expanded the Methods section (4.4) and the Discussion as limitation to explicitly address this point and provide the necessary rationale and references.
- In most in vitro experiments, the specific treatment conditions for each experimental group (including timing, sequence of Aβ/LPS exposure, and extract/compound administration) are not described in sufficient detail in either the Methods section or the figure legends, making it difficult to draw reliable conclusions from the results.
Response:
We sincerely thank the reviewer for this critical comment. We agree that precise description of treatment protocols is essential for experimental interpretation and reproducibility. We apologize for this oversight and have thoroughly revised Methods section 4.4 and all relevant figure legends to provide detailed, step-by-step treatment conditions for every experimental group. Additionally, we have included a comprehensive schematic timeline of the co-culture and treatment procedure (Figure S1) in the supplementary materials to enhance methodological clarity.
- The lack of explicit information on replicates, number of independent experiments, and exact controls for each assay makes it difficult to assess the robustness and reproducibility of the results.
: We thank the reviewer for this important comment. We have now added explicit experimental design details to Methods section 4.4 and a new section 4.8 (Statistical Analysis) that specify:
Replication: n=4 technical replicates per condition all assays
Control groups: Three standardized control groups (Normal Control, Control for the disease model, Positive Control) explicitly defined for each assay
Statistical framework: Enhanced statistical analysis section with detailed sample sizes and analysis methods
- Detailed experimental protocols should be provided for all assays, including cell differentiation procedures, co-culture setup, and treatment regimens for both PC12 and Caco-2 compartments.
: We have substantially expanded Methods section 4.4 to include: 1) Detailed 23-day timeline: Step-by-step protocol from cell expansion through sample collection (Supplementary Figure S1: Added comprehensive experimental timeline schematic) 2) Specific differentiation procedures: NGF treatment schedule for PC12 cells and TEER validation for Caco-2 monolayers 3) Treatment sequence: Sequential inflammatory challenge (Aβ₂₅₋₃₅ + LPS) followed by compound treatment with exact timing and compartment specifications 4) Quality control measures: Microscopic monitoring, TEER measurements, and contamination testing protocols
- While the network pharmacology analysis identified several key targets (AKT1, IL-1β, IL-6) and pathways (e.g., AGE–RAGE, PI3K–Akt), many of these are well-documented in the AD literature. The manuscript should clearly articulate what is novel about these findings relative to existing studies.
Response:
We thank the reviewer for this insightful comment, which allows us to better highlight the novel aspects of our work. We fully agree that individual targets like AKT1, IL-1β, and IL-6 are well-established in AD pathology. The novelty of our findings does not lie in the identification of new targets, but rather in the novel multi-target, multi-system approach through which traditional Chinese medicine (TCM) extracts simultaneously modulate these well-known targets across both the central nervous and peripheral gastrointestinal systems via the gut-brain axis.
From Single-Target to Multi-Target Synergy: While previous studies often focus on modulating a single target (e.g., just AKT or just IL-6 in neurons), our network pharmacology-guided approach demonstrates that VT, PM, AVF, and EF extracts naturally engage a network of AD-related targets simultaneously. This multi-target action is a fundamental principle of TCM and may offer superior efficacy by addressing the complex, multifactorial nature of AD, potentially overcoming the limitations of single-target drug failures.
From Isolated CNS to Integrated Gut-Brain Axis Modulation: The primary novelty is the demonstration that these extracts exert dual protective effects on both Aβ-stressed neurons and LPS-inflamed intestinal epithelial cells within the same experimental system. Most prior studies on these pathways are confined to neuronal models. We show for the first time that these extracts can: Attenuate neuroinflammation, ameliorate intestinal inflammation, preserve intestinal barrier integrity, reduce systemic inflammatory cues and improve mitochondrial function in both cell types. It was discussed in page 17-18.
We have now revised the Discussion section to more clearly and explicitly articulate these points of novelty.
- The connection between KEGG/GO pathway enrichment results and the specific in vitro experiments is not sufficiently developed. The authors should explicitly link which predicted pathways were validated experimentally.
Response: AGE-RAGE Signaling Pathway in Diabetic Complications (hsa04933): This pathway was also significantly enriched. We experimentally validated its involvement by demonstrating that the extracts significantly downregulated the mRNA expression of key inflammatory cytokines (TNF-α, IL-1β, IL-6) that are core effectors of the AGE-RAGE axis, in both PC12 and Caco-2 cells (Fig. 4A,4B,4C,4D).
Cytokine Activity (GO:0005125) & Inflammatory Response: The GO analysis highlighted cytokine activity. This was directly validated by our qPCR results showing a reduction in TNF-α, IL-1β, and IL-6 and by the reduction in mast cell numbers shown by toluidine blue staining (Fig. 6A, B).
We have amended the manuscript to ensure these links are unmistakably clear, Results Section 2.1 and 2.5 (Network pharmacology analysis). It was explained in the discussion (page 18).
- The leap from in vitro findings to potential clinical application is premature without in vivo validation and pharmacokinetic data.
Response:
We thank the reviewer for this critical and valid comment. We completely agree that the leap from in vitro findings to clinical application requires rigorous in vivo validation and comprehensive pharmacokinetic studies. We acknowledge that our in vitro co-culture model, despite its sophistication, cannot fully recapitulate whole-organism complexity including drug absorption, distribution, metabolism, and integrated physiological responses.
We would like to clarify that the primary objective of this study was to elucidate the fundamental cellular and molecular mechanisms of these herbal extracts using a physiologically relevant co-culture system, thereby providing mechanistic insights that can guide future in vivo investigations. We present this work as an essential foundational study that establishes proof-of-concept for dual gut-brain protection mechanisms.
I n response to this important concern, we have revised our manuscript to remove any language suggesting immediate clinical applicability and have repositioned our findings as mechanistic groundwork requiring extensive preclinical validation. We have also expanded our limitations section (page 19) to explicitly acknowledge the need for comprehensive in vivo studies, pharmacokinetic profiling, safety assessment, and dose optimization before any clinical translation can be considered.
- Several figures (e.g., network diagrams in Fig. 1B/C) lack clarity, and the figure legends do not provide enough detail for the reader to fully understand the experimental setup and results.
Response:
We thank the reviewer for this valuable feedback, which has helped us significantly improve the clarity and interpretability of our figures. We agree that the complexity of network pharmacology figures can be challenging to decipher without detailed guidance. We have taken the following steps to address this concern:
Enhanced Figure Legends: We have completely rewritten and expanded the legends for Figures 1, 2, 3, 4, 5, and 6 to provide a comprehensive, stand-alone description of the experimental setup, what each panel represents, and how to interpret the key findings. The legends now explicitly define all symbols, abbreviations, and statistical details.
Improved Figure Quality: We have re-generated the network diagrams in Figure 1 to improve visual clarity. This includes: Increasing the font size for key labels to enhance readability.
- Statistical annotations should be standardized, and all abbreviations should be defined upon first use in each figure legend.
Response:
We thank the reviewer for this meticulous comment, which has helped us improve the clarity and professionalism of our data presentation. We have thoroughly reviewed all figure and table legends throughout the manuscript and have made the following comprehensive revisions to ensure full compliance:
Standardized Statistical Annotations: We have standardized the statistical representation across all figures. Different lowercase letters (a, b, c, d) are now consistently used to denote statistically significant differences between groups (p < 0.05) as determined by one-way ANOVA followed by Tukey's post-hoc test. Groups that share a common letter are not significantly different from each other. This system has been applied uniformly to all bar graphs.
Definition of All Abbreviations: In every figure and table legend, all abbreviations are now explicitly defined upon their first use within that specific legend. This ensures that each legend is complete and can be understood independently, without requiring the reader to cross-reference the main text or other legends.
- While the introduction cites recent work on gut–brain axis involvement in AD, the discussion does not adequately compare the present findings with other in vitro and in vivo studies using similar herbal extracts or co-culture systems.
Response:
We thank the reviewer for this insightful comment. We added some discussion about gut-brain axis in page 18-19.
- Recent in vivo studies investigating gut–brain axis modulation in AD should be incorporated to strengthen the translational relevance of the work.
Response:
We agreed with the reviewer that in vivo validation is paramount for establishing the translational relevance of any in vitro findings, particularly for a complex system like the gut-brain axis. We thank the reviewer for this important suggestion.
In direct response to this point, we have now revised the Discussion section to incorporate and discuss key recent in vivo studies that have demonstrated the efficacy of gut-brain axis modulation in AD models. This places our in vitro mechanistic findings within the broader context of compelling in vivo research, strengthening the rationale for our approach.
The therapeutic potential of targeting the gut-brain axis in AD is strongly supported by a growing body of in vivo evidence. Recent studies have demonstrated that modulating gut microbiota or improving intestinal barrier function can attenuate neuroinflammation and cognitive decline in various AD rodent models. For instance, modulation of the gut microbiota via parasympathetic inhibition improved memory function and reduced Aβ deposition.
Our in vitro findings provide mechanistic cellular-level insights that align with these observed in vivo benefits. The ability of VT, PM, AVF, and EF extracts to simultaneously enhance neuronal viability, suppress neuroinflammation, and fortify intestinal barrier integrity in our co-culture system provides a mechanistic foundation for understanding the multi-system improvements reported in animal studies. This convergence between our mechanistic findings and successful in vivo interventions supports the translational potential of our proposed gut-brain axis therapeutic approach.
We added the translational relevance in vivo study in the discussion section (page 17-18).
Reviewer 2 Report
Comments and Suggestions for Authors
A brief summary
This article investigates the neuroprotective and anti-inflammatory effects of four traditional Chinese medicine plant extracts in Alzheimer's disease. The authors employed a comprehensive approach combining network pharmacology with experimental validation using a co-culture model of PC12 cells (neurons) and Caco-2 cells (intestinal epithelial cells). The study demonstrated that extracts of Vitex trifolia, Plantago major, Apocyni Veneti Folium, and Eucommiae folium effectively activate the PI3K-Akt-GSK-3β signaling pathway and reduce oxidative stress and inflammation in both neuronal and intestinal cells.
This work will be valuable for researchers studying neurodegenerative diseases, traditional medicine specialists, and developers of new therapeutic approaches for Alzheimer's disease treatment. The novelty of the study lies in applying an innovative co-culture model for simultaneous investigation of gut-brain axis effects, which has not been previously used in traditional Chinese medicine research for Alzheimer's disease.
General concept comments
- It is unclear why donepezil hydrochloride was specifically chosen as a control. There is no justification for selecting this particular drug. In this case, it is unclear how to evaluate the effect of the identified compounds compared to other modern approaches for treating Alzheimer's disease. Adding the most common ones would make the article too lengthy. However, it is essential to mention why this specific drug was chosen and not other treatment methods.
- There is a lack of clear distinction between the effects of individual bioactive compounds and whole extracts, which complicates understanding of the mechanisms of action. A systematic comparison of the effectiveness of individual compounds with combined preparations should be included to identify synergistic effects. Bioactive compounds do not necessarily have to have positive effects. It is possible that when combined, they exhibit antagonistic and synergistic actions.
Specific comments
- All images need resolution enhancement. Currently, all images are blurry, indistinct, making it impossible to discern fine details.
- Fig. 1C. The nodes of the protein-protein interaction map do not carry informational value, yet protein structures are visualized within them. In String settings, the display of protein structures in nodes can be removed. Gene labels should be moved inside the nodes because currently everything merges with the graph edges.

Author Response
Replies to the reviewers’ comments
We appreciate the valuable comments and suggestions for our paper. We made a sincere effort to address each comment and make appropriate revisions. We have point-by-point replies to each comment.
General concept comments
- It is unclear why donepezil hydrochloride was specifically chosen as a control. There is no justification for selecting this particular drug. In this case, it is unclear how to evaluate the effect of the identified compounds compared to other modern approaches for treating Alzheimer's disease. Adding the most common ones would make the article too lengthy. However, it is essential to mention why this specific drug was chosen and not other treatment methods.
Response:
We thank the reviewer for this pertinent comment. We agree that the rationale for selecting donepezil hydrochloride as the positive control should have been explicitly stated. Donepezil was chosen for the following compelling reasons:
Donepezil hydrochloride is a first-line, FDA-approved acetylcholinesterase inhibitor (AChEI) for the symptomatic treatment of mild to moderate Alzheimer's disease. It is one of the most widely used and clinically established drugs for AD, making it a highly relevant benchmark for evaluating potential therapeutic agents.
A central pathological feature of AD is the cholinergic deficit. Our study specifically measured acetylcholinesterase (AChE) activity as a key endpoint (Table 1). Donepezil, as a potent and selective AChEI, serves as a direct and mechanistically appropriate positive control for this specific readout, allowing us to directly compare the extract's anti-AChE efficacy against a clinically proven standard.
We added the donepezil as positive control in the method section with reference in page 2.
(Supplementing in Introduction)
- There is a lack of clear distinction between the effects of individual bioactive compounds and whole extracts, which complicates understanding of the mechanisms of action. A systematic comparison of the effectiveness of individual compounds with combined preparations should be included to identify synergistic effects. Bioactive compounds do not necessarily have to have positive effects. It is possible that when combined, they exhibit antagonistic and synergistic actions.
Response:
We thank the reviewer for this insightful comment regarding the distinction between individual bioactive compounds and whole extracts. We acknowledge that our study design does not provide direct systematic comparisons between individual compounds and their parent extracts, which limits our ability to identify potential synergistic or antagonistic interactions.
In our experimental approach, individual bioactive compounds (baicalein, kaempferol, luteolin, and quercetin) were tested as representative active constituents to understand their individual mechanisms of action, while whole herbal extracts (VT, PM, AVF, and EF) were evaluated as complex mixtures containing multiple bioactive compounds along with other phytochemicals. However, we did not conduct direct head-to-head comparisons between equivalent concentrations of individual compounds versus their combinations or parent extracts, which would be necessary to definitively identify synergistic, additive, or antagonistic effects.
We agree that bioactive compounds in complex herbal preparations may exhibit various interaction patterns, including antagonistic effects that could potentially reduce overall therapeutic efficacy. This represents an important limitation of our current study design and should be addressed in future research through systematic dose-response studies comparing individual compounds, defined combinations, and whole extracts using appropriate mathematical models (such as combination index analysis) to quantify interaction effects.
Specific comments
- All images need resolution enhancement. Currently, all images are blurry, indistinct, making it impossible to discern fine details.
: We increased the resolution of the images.
- Fig. 1C. The nodes of the protein-protein interaction map do not carry informational value, yet protein structures are visualized within them. In String settings, the display of protein structures in nodes can be removed. Gene labels should be moved inside the nodes because currently everything merges with the graph edges.
: We revised the figure 1C according to the reviewer’s comment.
Round 2
Reviewer 1 Report
Comments and Suggestions for Authors
My suggestions are adequately handled in the revised MS.